# Perioperative Complications of Anterior Decompression with Fusion in Degenerative Cervical Myelopathy—A Comparative Study between Ossification of Posterior Longitudinal Ligament and Cervical Spondylotic Myelopathy Using a Nationwide Inpatient Database

**DOI:** 10.3390/jcm11123398

**Published:** 2022-06-13

**Authors:** Shingo Morishita, Toshitaka Yoshii, Hiroyuki Inose, Takashi Hirai, Yu Matsukura, Takahisa Ogawa, Kiyohide Fushimi, Junya Katayanagi, Tetsuya Jinno, Atsushi Okawa

**Affiliations:** 1Department of Orthopedic Surgery, Tokyo Medical and Dental University Graduate School of Medicine, 1-5-45 Yushima, Bunkyo-ku, Tokyo 113-8519, Japan; morsorth@tmd.ac.jp (S.M.); inose.orth@tmd.ac.jp (H.I.); hirai.orth@tmd.ac.jp (T.H.); matsukura.orth@tmd.ac.jp (Y.M.); takahisa.o@gmail.com (T.O.); okawa.orth@tmd.ac.jp (A.O.); 2Department of Orthopedic Surgery, Dokkyo Medical University Saitama Medical Center, 2-1-50 Minamikoshigaya, Koshigaya-shi, Saitama 343-8555, Japan; junya@dokkyomed.ac.jp (J.K.); jinnot@dokkyomed.ac.jp (T.J.); 3Department of Health Policy and Informatics, Tokyo Medical and Dental University Graduate School of Medicine, 1-5-45 Yushima, Bunkyo-ku, Tokyo 113-8519, Japan; kfushimi.hci@tmd.ac.jp

**Keywords:** perioperative complications, costs, cervical ossification of the longitudinal ligament, cervical spondylotic myelopathy, anterior decompression with fusion, diagnosis procedure combination database

## Abstract

For ossification of the posterior longitudinal ligament (OPLL) and cervical spondylotic myelopathy (CSM), anterior decompression with fusion (ADF) can accurately achieve spinal decompression. However, the difference in perioperative complications in ADF between OPLL and CSM is poorly described. This study aimed to investigate the perioperative complication rates of patients with degenerative cervical myelopathy undergoing ADF, represented by OPLL and CSM, using a large national inpatient database. In the OPLL and CSM groups, postoperative complication (systemic and local) rates, reoperation rates, medical costs during hospitalization, and mortality after propensity score matching were compared. After matching, 1197 matched pairs were made. The incidence of total systemic complications was similar between both groups (OPLL, 12.4%; CSM, 12.2%). In the OPLL group, more perioperative local complications (cerebrospinal fluid leakage: CSFL, [OPLL, 2.7%; CSM, 0.3%] and surgical site infection: SSI [OPLL, 2.1%; CSM, 0.9%]) were detected, and the hospitalization cost was approximately $3200 higher than that in the CSM group. Moreover, medical costs were significantly higher in patients who experienced complications in both OPLL and CSM. The frequency of perioperative complications of OPLL and CSM in ADF was detailed using large real-world data. Compared to CSM, OPLL had more perioperative complications such as CSFL and SSI, and higher medical costs. Regardless of the disease, medical costs were significantly increased when perioperative complications occurred.

## 1. Introduction

Degenerative cervical myelopathy (DCM) is a general term for symptomatic spinal cord compression caused by degenerative changes in the cervical spine [1]. The main diseases included in DCM are ossification of the posterior longitudinal ligament (OPLL) and cervical spondylotic myelopathy (CSM) [2]. OPLL is often detected in Asian countries, and heterotopic bone formation with OPLL causes nerve compression when the ossification foci are enlarged [3]. OPLL has also been linked to high body mass index (BMI) and metabolic diseases (e.g., diabetes) [4]. In contrast, CSM is a condition associated with neurological deterioration caused by spinal cord compression due to degenerative disk changes, osteophyte formation, and ligamentum flavum hypertrophy [5]. Although both diseases generally cause compressive spinal cord myelopathy, the patient backgrounds differ between OPLL and CSM.

As clinical symptoms progress in patients with DCM, conservative treatment may become ineffective; therefore, surgery is preferred in such patients [6]. Anterior decompression with fusion (ADF) is one of the treatment options for OPLL and CSM when the spinal cord is compressed from the front [7,8]. Anterior cervical approach surgery is considered a fundamental and sensible technique that allows for the direct relief of spinal cord compression [9]. In patients with both OPLL and CSM, ADF effectively improves neurological deficits and activity levels, as reported in several studies [10,11].

In general, surgery via the anterior approach for cervical DCM is challenging, particularly for OPLL, and perioperative complications are common [12,13]. The differences between OPLL and CSM were examined in few studies with a focus on perioperative complications after ADF even though several reviews have compared anterior and posterior surgery in both OPLL and CSM [14,15]. Therefore, using a nationwide inpatient database, this study aimed to determine the perioperative complication details by comparing post-ADF complication rates and medical costs between the two different diseases, OPLL and CSM. In addition, patients’ backgrounds were strictly matched to compare the diseases without biases.

## 2. Materials and Methods

### 2.1. Data Source and Study Population

We obtained all data for the current study from the Diagnosis Procedure Combination (DPC) database, which is a nationwide database of hospitalized patients in Japan [16,17]. Age, sex, BMI, smoking status, scheduled or unscheduled admission, emergency, academic or nonacademic hospital, scores of activities of daily living (ADL), preoperative comorbidities, perioperative complications according to ICD-10 codes, surgical procedure, reoperation due to complication, transfusion, cost, and mortality during hospitalization were all recorded in the database. The study included hospitalized patients who underwent ADF (including anterior cervical decompression and fusion: ACDF and anterior cervical corpectomy and fusion: ACCF) between 1 April 2010 and 31 March 2016, with a principal diagnosis of OPLL or CSM. Patients who underwent ADF but were diagnosed with spinal cord injury, cervical radiculopathy, cervical disk herniation, and cervical amyotrophy and those diagnosed with OPLL or CSM but who underwent combined anterior and posterior surgery were excluded from the study.

### 2.2. Variables and Outcomes

Data on preoperative comorbidities and perioperative complications were obtained from the database. Preoperative comorbidities include diabetes mellitus, cardiovascular diseases, cerebrovascular diseases, respiratory diseases, renal failure, hepatic failure, gastric ulcer and hemorrhage, malignant tumor, rheumatoid arthritis, and osteoporosis. Perioperative complications include systemic complications (i.e., cardiovascular disease, cerebrovascular disease, respiratory failure, pneumonia, dysphagia, renal failure, hepatic failure, deep venous thrombosis, pulmonary embolism, sepsis, and delirium) and local complications (i.e., surgical site infection, paralysis, meningitis, spinal fluid leakage, and hematoma). Furthermore, reoperations for perioperative systemic complications (cardiac, cerebral, gastric complications, inferior vena cava filter placement, tracheotomy, and shunt formation) and local complications (treatment for wound and debridement) were included. The perioperative complications, reoperation rates, inpatient medical costs, mortality, and ADL scores at discharge were compared between patients with OPLL and CSM after propensity score matching. Surgical site infection in this study was defined as an infection that occurs after surgery in the part of the body where the surgery took place during the hospitalization according to Centers for Disease Control and Prevention (CDC) definition. Surgical site infection in this study were defined as those that occurred during the hospital stay, not within 30 days after surgery. Regarding the reoperation, closure for fluid leakage was defined as suture of the dura mater, and treatment for wound trouble was defined as re-suture at the superficial (skin and subcutaneous) layer of the wound. Patients were then further divided into two groups according to the presence or absence of complications (systemic, local, or total), and medical costs were compared.

### 2.3. Statistical Analysis

Propensity scores were calculated using logistic regression analysis with the known preoperative items as explanatory variables [18,19,20]. Explanatory variables for logistic regression included age, sex, BMI, smoking, admission type, emergency, hospital type, ADL score, and preoperative comorbidities. In the OPLL and CSM groups, propensity score matching was performed using a caliper with a width of <0.2. After forming one-to-one pairs using matched patient backgrounds, a cohort was created and adapted to selection bias between the OPLL and CSM groups. Univariate analysis was first performed using raw data to examine patient backgrounds between the OPLL and CSM groups. Following the raw data analysis, univariate analysis was performed for patients matched by propensity scores to examine perioperative complication rates, reoperation rates, medical costs, mortality, and ADL scores. The complication rates for OPLL and CSM were also compared by age group (≥70 and <70). Furthermore, the length of hospital stay and medical costs in patients who developed complications were also examined by age (≥70 and <70). We performed an additional analysis for cases without perioperative complications. Categorical variables were analyzed using Fisher’s exact test or chi-square test, and continuous variables were compared using a Student’s *t*-test. Statistical analyses were performed using Stata IC version 16 (StataCorp, College Station, TX, USA). Statistical significance was set at a *p* value of <0.05.

## 3. Results

A total of 2993 patients were included: OPLL (*n* = 1333) and CSM (*n* = 1660; Table 1). Before propensity score matching, patients in the OPLL group were younger than those in the CSM group (OPLL vs. CSM [aged 60.3 vs. 62.6 years]; *p* < 0.001). Furthermore, the OPLL group included patients with higher BMI (25.2 vs. 23.8 kg/m^2^; *p* < 0.001), higher scheduled admission (93.3% vs. 91.0%; *p* = 0.018), and lower emergency transport (1.4% vs. 2.8%; *p* = 0.012) than the CSM group. In contrast, patients in the CSM group were associated with more preoperative comorbidities, such as cardiovascular diseases (OPLL vs. CSM: 4.5% vs. 6.3%; *p* = 0.030), respiratory diseases (0.5% vs. 1.3%; *p* = 0.037), renal failure (1.0% vs. 2.8%; *p* < 0.001), malignant tumor (0.6% vs. 1.5%; *p* = 0.025), and osteoporosis (1.6% vs. 3.1%; *p* = 0.008). Other comorbidities, including diabetes mellitus, were not significantly different between the two groups. After the one-to-one matching using propensity scores, 1197 pairs (2394 patients) were obtained and analyzed. Thus, the demographic characteristics of these 1197 pairs in the OPLL and CSM groups were well matched (Table 1).

After matching the OPLL and CSM groups, the perioperative systemic complication rates are displayed in Table 2. The total incidence of systemic complications was 12.4% in OPLL and 12.2% in CSM, respectively, without a significant difference. Between the OPLL and CSM groups, the incidence of respiratory problems, including pneumonia (OPLL vs. CSM [0.9% vs. 1.1%]; *p* = 0.68) and dysphagia (2.2% vs. 2.1%; *p* = 0.89) was also similar. For systemic complications, the reoperation rates were 0.9% and 0.8% in the OPLL and CSM groups, respectively, without difference between the two groups. The difference was not statistically significant, although in the OPLL group, perioperative blood transfusion tended to be higher (OPLL vs. CSM [7.4% vs. 5.8%]; *p* = 0.08; Table 2).

Table 3 demonstrates the local complications and reoperation rates for local complications after matching between the two groups. The incidence of local complications was higher in the OPLL compared to the CSM group (OPLL vs. CSM [6.1% vs. 2.9%], *p* < 0.001). The surgical site infection incidence (OPLL vs. CSM [2.1% vs. 0.9%]; *p* = 0.019) and spinal fluid leakage (2.7% vs. 0.3%; *p* < 0.001) were significantly higher in the OPLL compared to the CSM group. Moreover, in the OPLL group, fluid leakage closure (OPLL vs. CSM [0.4% vs. 0%]; *p* = 0.025), wound trouble treatment (1.8% vs. 0.8%; *p* = 0.027), and local complication reoperation (2.2% vs. 0.8%; *p* = 0.004) were more frequently needed (Table 3).

In addition, we compared patients ≥70 and those <70 in each disease (OPLL, CSM). Some of the systemic complications such as dysphagia, sepsis, and delirium showed significantly higher incidences in the ≥70 group for OPLL patients, but no differences were found for CSM patients. (Appendix A). There was no difference in local complications between patients ≥70 and <70 years for both OPLL and CSM patients.

Table 4 shows the in-hospital medical costs after propensity score matching. In the OPLL group, the costs were almost $3200 higher (OPLL vs. CSM [$23,164 vs. $19.993]; *p* < 0.001). Mortality and ADL scores at discharge were not significantly different between the two groups (Table 4).

Table 5 compares the in-hospital medical costs in each OPLL and CSM group after matching, dividing them into two groups with or without complications. Either in OPLL or the CSM group, the costs were significantly higher in the group with complications. For patients with OPLL, the cost difference between the patients with and without complications was about $5200 ($26,920 vs. $21,714; *p* < 0.001) and $4000 ($26,115 vs. $22,113; *p* = 0.002) for systemic and local complications, respectively. Furthermore, for patients with CSM, the cost difference between the patients with and without complications was about $6500 ($25,013 vs. $18,502; *p* < 0.001) for systemic complications (Table 5). In cases without complications, the length of hospital stay was significantly longer in OPLL cases than in CSM cases by about three to four days (Appendix A). The medical costs were higher in OPLL cases than in CSM cases even when complications did not occur.

## 4. Discussion

We focused on the comparison of different diseases, OPLL and CSM, in terms of perioperative complications and medical costs for ADF and to analyze them in a large national database. The comparison was performed under the strict matching of patients’ backgrounds between the OPLL and CSM. Japan is one of the countries with most aged individuals worldwide; therefore, a higher number of older patients (OPLL, 60.3 years; CSM, 62.6 years) were observed in the database used in this study than in a similar database study on anterior cervical spine surgery conducted in Western countries (aged 55 years) [21]. Thus, the present study results will provide important information that surgeons should focus on when surgical operations are performed on elderly patients. Patients in the current database had different backgrounds because those with CSM were older, had a lower mean BMI, and had a higher preoperative comorbidity rate than those with OPLL (e.g., cardiovascular disease, respiratory disease, renal failure, malignancy, and osteoporosis). Therefore, using propensity scores, the patient’s backgrounds were strictly matched between the OPLL and CSM groups to assess ADF perioperative complication rates (Table 1) accurately.

The incidence of perioperative systemic complications was not significantly different between the two groups. However, when compared with the database study conducted in the USA (incidence of perioperative complications, 8.9%), the incidence of perioperative complications in the present study was relatively high (OPLL, 12.4%; CSM, 12.2%) [22]. Among them, high incidences were observed in cardiovascular complications (OPLL, 2.7%; CSM, 2.9%) and dysphagia (OPLL, 2.2%; CSM, 2.1%). A study on cardiac complications after cervical spine surgery found that age, multilevel spinal fusion, and bleeding were risk factors for postoperative cardiac events [23]. Anterior cervical surgery occasionally requires multilevel fusion, particularly in patients with OPLL, resulting in increased blood loss and transfusion [24]. According to the data collected in this study, patients were relatively older and had more transfusions in both the OPLL and CSM groups, which may cause perioperative cardiovascular stress and lead to various cardiac events. During surgery, continued medial retraction of the esophagus and pharynx is also required in the anterior cervical spine approach [25]. Esophageal irritation, laryngeal nerve injury, and edema of the posterior pharyngeal wall can cause dysphagia [26]. The incidence of postoperative dysphagia in anterior surgery was >10 times higher than that in posterior surgery [12]. Our data showed that the incidence of dysphagia was significantly higher in the ≥70 group for OPLL patients, but no differences were found for CSM patients. Thus, after anterior cervical spine surgery, careful attention should be given to dysphagia, especially for elderly OPLL patients.

This study showed that after anterior cervical spine surgery, the incidence of cerebrospinal fluid (CSF) leakage in OPLL was approximately nine times higher than that in CSM. No previous studies have shown a higher incidence of CSF leakage in OPLL in direct comparison with CSM. A report stated that the incidence of dural tears and CSF leakage was at most 8% after anterior cervical corpectomy for all pathologies of the cervical spine [27]. However, regarding cervical OPLL, several studies have reported that after anterior cervical surgery, the incidence of CSF leakage is exceptionally high, ranging from 6.7% to 53.3% [28,29,30,31,32]. Consequently, ADF for OPLL causes CSF leakage because of the resection of the ossified dura and/or adhesion detachment [33]. CSF leakage after ADF for OPLL can be challenging and may cause pseudomeningocele, dyspnea, cutaneous CSF leakage fistula, and meningitis [34]. In this study, the frequency of surgical site infection in OPLL was more than twice than that in CSM. However, no cases of deep surgical site infection that required debridement were noted in this cohort. CSF leakage outside the skin and prolonged wound healing may contribute to the increased risk of infection in patients with OPLL. As previously reported, anterior decompression using the floating technique or anterior controllable antidisplacement and fusion (ACAF) instead of resecting OPLL may effectively reduce the risk of CSF leakage and related complications [9,35].

Postoperative complications were more frequently observed in patients with OPLL, resulting in higher medical costs. Medical costs during hospitalization were approximately $3200 higher in the OPLL group than in the CSM group (OPLL vs. CSM [$23,164 vs. $19,993]; *p* < 0.001; Table 4). Several previous studies have shown that perioperative complications in spine surgeries lead to increased medical costs [36,37]. The incremental cost of treating severe perioperative complications has been noted to exceed $20,000 [36]. In this cohort, perioperative complications increased hospitalization costs for patients with OPLL and CSM (Table 5). In particular, perioperative complications increased the medical costs in the OPLL and CSM groups by approximately $4000–$5000 and $2000–$6000, respectively. When perioperative complications were noted in the OPLL group, the total medical costs exceeded $26,000. In OPLL or CSM, reducing perioperative complications by devising surgical techniques is critically important. It is possible that the longer hospital stay may have an impact on higher costs.

### Limitations

This study has several limitations. Because the DPC database only contained information during hospitalization, we could not consider mortality, postoperative complications, and reoperation after discharge. This is especially likely to affect complication rates for patients discharged earlier than 30 days, the Center of Disease Control and Prevention’s definition of surgical site infection [38]. Additionally, some clinically important findings could not be reflected in the database system because of the lack of images (including the canal occupancy ratio of OPLL) and laboratory data. For example, the database excluded some data, such as details of the surgical procedure (ACDF, ACCF or ACAF), number of surgical levels, details of death, severity of neurological symptoms, patients’ pathology, cervical spine alignment, date of complication diagnosis, and wound condition.

Despite the limitations, this study’s findings provide valuable information to surgeons and patients with DCM to know the characteristics that lead to higher complication rates and higher medical costs. In addition, it is necessary to construct a database containing detailed clinical and imaging findings in future studies.

## 5. Conclusions

We have detailed the frequency of perioperative complications of OPLL and CSM in ADF using large real-world data. An unbiased comparison using propensity score matching revealed that OPLL was clearly associated with more perioperative complications (i.e., CSF leakage and surgical site infection) and higher medical costs than CSM. Medical costs were significantly increased when perioperative complications occurred in both the OPLL group and the CSM group.

## Figures and Tables

**Table 1 jcm-11-03398-t001:** Patient characteristics between OPLL and CSM before and after matching.

	Before Propensity Score-Matching			After Propensity Score-Matching		
	OPLL (*n* = 1333)	CSM (*n* = 1660)	*p* Value	OPLL (*n* = 1197)	CSM (*n* = 1197)	*p* Value
Age (years)	60.3 ± 11.3	62.6 ± 13.0	<0.001 ***	61.3 ± 10.9	60.8 ± 12.9	0.35
Sex			0.60			0.89
Male	948 (71.1%)	1166 (70.2%)		854 (71.4%)	857 (71.6%)	
Female	385 (28.9%)	494 (29.8%)		343 (28.6%)	340 (28.4%)	
BMI (kg/m^2^)	25.2 ± 4.2	23.8 ± 3.9	<0.001 ***	24.4 ± 3.2	24.7 ± 3.7	0.11
Smoking			0.99			0.41
Yes	682 (51.2%)	849 (51.1%)		615 (51.4%)	635 (53.0%)	
No	651 (48.8%)	811 (48.9%)		582 (48.6%)	562 (47.0%)	
Admission type			0.018 *			0.33
Scheduled	1244 (93.3%)	1510 (91.0%)		1111 (92.8%)	1123 (93.8%)	
Unscheduled	89 (6.7%)	150 (9.0%)		86 (7.2%)	74 (6.2%)	
Emergency transport			0.012 *			0.61
Yes	19 (1.4%)	46 (2.8%)		19 (1.6%)	16 (1.3%)	
No	1313 (98.6%)	1614 (97.2%)		1178 (98.4%)	1181 (98.7%)	
Hospital type			0.99			0.96
Academic	243 (18.2%)	303 (18.2%)		215 (18.0%)	214 (17.9%)	
Non-academic	1090 (81.8%)	1357 (81.8%)		982 (82.0%)	983 (82.1%)	
ADL score for admission (points)	17.3 ± 5.0	17.0 ± 5.5	0.23	17.2 ± 5.1	17.3 ± 5.2	0.75
Preoperative comorbidities						
Diabetes mellitus	296 (22.2%)	334 (20.1%)	0.16	253 (21.1%)	255 (21.3%)	0.92
Cardiovascular disease	60 (4.5%)	105 (6.3%)	0.030 *	57 (4.8%)	65 (5.4%)	0.46
Cerebrovascular disease	31 (2.3%)	46 (2.8%)	0.44	31 (2.6%)	27 (2.3%)	0.60
Respiratory disease	7 (0.5%)	21 (1.3%)	0.037 *	7 (0.6%)	7 (0.6%)	>0.99
Renal failure	13 (1.0%)	46 (2.8%)	<0.001 ***	12 (1.0%)	12 (1.0%)	>0.99
Hepatic failure	39 (2.9%)	53 (3.2%)	0.67	38 (3.2%)	36 (3.0%)	0.81
Gastric ulcer and hemorrhage	48 (3.6%)	64 (3.9%)	0.72	43 (3.6%)	43 (3.6%)	>0.99
Malignancy	8 (0.6%)	24 (1.5%)	0.025 *	8 (0.7%)	4 (0.3%)	0.25
Rheumatoid arthritis	3 (0.2%)	12 (0.7%)	0.06	3 (0.3%)	8 (0.7%)	0.13
Osteoporosis	21 (1.6%)	51 (3.1%)	0.008 **	20 (1.7%)	18 (1.5%)	0.74

Data were presented as *n* (%) or mean ± SD. Significant values are given as follows. * *p* < 0.05, ** *p* < 0.01, *** *p* < 0.001. OPLL, ossification of posterior longitudinal ligament; CSM, cervical spondylotic myelopathy; SD, standard deviation; BMI, body mass index; ADL, activities of daily living.

**Table 2 jcm-11-03398-t002:** Systemic complications, reoperation, and blood transfusion after matching.

	After Propensity Score-Matching		
	OPLL (*n* = 1197)	CSM (*n* = 1197)	*p* Value
Systemic complications			
At least one complication	148 (12.4%)	146 (12.2%)	0.90
Cardiovascular disease	32 (2.7%)	35 (2.9%)	0.71
Cerebrovascular disease	9 (0.8%)	9 (0.8%)	>0.99
Respiratory failure	13 (1.1%)	16 (1.3%)	0.58
Pneumonia	11 (0.9%)	13 (1.1%)	0.68
Dysphagia	26 (2.2%)	25 (2.1%)	0.89
Renal failure	3 (0.3%)	0 (0%)	0.08
Hepatic failure	2 (0.2%)	6 (0.5%)	0.16
Deep venous thrombosis	3 (0.3%)	7 (0.6%)	0.21
Pulmonary embolism	1 (0.1%)	0 (0%)	0.32
Sepsis	2 (0.2%)	6 (0.5%)	0.16
Delirium	3 (0.3%)	3 (0.3%)	>0.99
Reoperation			
At least one operation	11 (0.9%)	9 (0.8%)	0.65
Cardiac	1 (0.1%)	2 (0.2%)	0.56
Cerebral	0 (0%)	2 (0.2%)	0.16
Gastric (including endoscope)	6 (0.5%)	1 (0.1%)	0.06
Vena cava filter	0 (0%)	0 (0%)	NA
Tracheoplasty or Tracheotomy	4 (0.3%)	3 (0.3%)	0.71
Shunt form	0 (0%)	1 (0.1%)	0.32
Blood transfusion	88 (7.4%)	67 (5.6%)	0.08

Data are presented as *n* (%). OPLL, ossification of posterior longitudinal ligament; CSM, cervical spondylotic myelopathy; NA, not applicable.

**Table 3 jcm-11-03398-t003:** Local complications and reoperation after matching.

	After Propensity Score-Matching		
	OPLL (*n* = 1197)	CSM (*n* = 1197)	*p* Value
Local complications			
At least one complication	73 (6.1%)	35 (2.9%)	<0.001 ***
Surgical site infection	25 (2.1%)	11 (0.9%)	0.019 *
Paralysis	14 (1.2%)	9 (0.8%)	0.30
Meningitis	2 (0.2%)	0 (0%)	0.16
Spinal fluid leakage	32 (2.7%)	4 (0.3%)	<0.001 ***
Hematoma	7 (0.6%)	12 (1.0%)	0.25
Reoperation			
At least one reoperation	26 (2.2%)	9 (0.8%)	0.004 **
Treatment for wound trouble	21 (1.8%)	9 (0.8%)	0.027 *
Closure for fluid leakage	5 (0.4%)	0 (0%)	0.025 *
Debridement	0 (0%)	0 (0%)	NA

Data are presented as *n* (%). Significant values are given as follows. * *p* < 0.05, ** *p* < 0.01, *** *p* < 0.001. OPLL, ossification of posterior longitudinal ligament; CSM, cervical spondylotic myelopathy; NA, not applicable.

**Table 4 jcm-11-03398-t004:** Cost, mortality, and ADL score for discharge after matching.

	After Propensity Score-Matching		
	OPLL (*n* = 1197)	CSM (*n* = 1197)	*p* Value
Cost (mean ± SD) ($)	23,164 ± 11,134	19,993 ± 10,629	<0.001 ***
Mortality	1 (0.1%)	2 (0.2%)	0.56
ADL score for discharge (points)	17.6 ± 4.5	17.8 ± 4.4	0.42

Data were presented as *n* (%) or mean ± SD. Significant values are given as follows. *** *p* < 0.001. OPLL, ossification of posterior longitudinal ligament; CSM, cervical spondylotic myelopathy; SD, standard deviation; ADL, activities of daily living.

**Table 5 jcm-11-03398-t005:** Cost between complication (+) and (−) in OPLL and CSM after matching.

OPLL (*n* = 1197)	Systemic Complication (+) (*n* = 148)	Systemic Complication (−) (*n* = 1049)	*p* Value
Cost ($)	26,920 ± 15,337	21,714 ± 9769	<0.001 ***
	Local complication (+) (*n* = 73)	Local complication (−) (*n* = 1124)	*p* Value
Cost ($)	26,115 ± 12,645	22,113 ± 10,572	0.002 **
	Total complication (+) (*n* = 209)	Total complication (−) (*n* = 988)	*p* Value
Cost ($)	26,701 ± 14,742	21,438 ± 9449	<0.001 ***
**CSM (*n* = 1197)**	**Systemic Complication (+) (*n* = 146)**	**Systemic Complication (−) (*n* = 1051)**	***p* Value**
Cost ($)	25,013 ± 18,793	18,502 ± 8120	<0.001 ***
	Local complication (+) (*n* = 35)	Local complication (−) (*n* = 1162)	*p* Value
Cost ($)	21,852 ± 8758	19,219 ± 10,294	0.13
	Total complication (+) (*n* = 177)	Total complication (−) (*n* = 1020)	*p* Value
Cost ($)	24,508 ± 17,514	18,392 ± 8063	<0.001 ***

Data were presented as mean ± SD. Significant values are given as follows. ** *p* < 0.01, *** *p* < 0.001. OPLL, ossification of posterior longitudinal ligament; CSM, cervical spondylotic myelopathy; SD, standard deviation.

## Data Availability

The datasets generated during and/or analyzed during the current study are available from the corresponding author on reasonable request.

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
