# Peer review of "Perioperative Complications of Anterior Decompression with Fusion in Degenerative Cervical Myelopathy—A Comparative Study between Ossification of Posterior Longitudinal Ligament and Cervical Spondylotic Myelopathy Using a Nationwide Inpatient Database"

_jcm, 2022, doi:10.3390/jcm11123398_

Round 1

Reviewer 1 Report

All my comments have been satisfactory adressed.

Reviewer 2 Report

I have no further issues. The manuscript can be accepted for publication

This manuscript is a resubmission of an earlier submission. The following is a list of the peer review reports and author responses from that submission.

Round 1

Reviewer 1 Report

Despite the authors cannot provide imaging data, the majority of my mentioned issues are now clarified. 

Reviewer 2 Report

Thank you for this revised version.

Several limitations of the manuscript are due to the lack of relevant data in the used database. The manuscript presents many interesting aspects of the problem but cannot fully explain the findings since data is missing.

I suggest that the limitation section is highlighted as “4.1 Limitations” and that all the limitations of the manuscript are clearly presented in that section.

·        Database does not differentiate between ACDF and ACCF

·        Number of levels treated is not stated in the database

·        Database only lists complications during initial hospital stay and not 30 days

·        Lack of data on mortality

·        Lack of data on disease severity

·        Lack of radiological data

·        What is the time frame covered for reoperations? Is it only during hospital stay?

Highlight the good results in elderly as a strength och the study. Maybe you could compare <70 with those >70? To show if there is a difference between age groups.

Abstract line 21: “However, the difference in perioperative systemic complications in ADF between OPLL and CSM is unknown.” Please use some other wording than unknown: poorly described, unsatisfactory investigated, a matter of discussion, conflicting, etc.. And remove “systemic” since you are investigating all complications.

As also pointed out by reviewer 2, the Abstract lacks a solid conclusion. What is the take home message? You could revise the conclusion paragraph at the end of the manuscript and move the content also to the abstract.

Discussion line 255: “Anterior decompression using the floating technique or anterior controllable antidisplacement technique instead of resecting OPLL may effectively reduce the risk of CSF leakage and related complications”. Could you see that in the database?